# Attenuating effects of inflammatory pathway by prolonged left ventricular unloading after myocardial infarction in male rats

Jingwen Gao[◐], Yasushige Shingu[◐]*, Ryota Azuma[¤], Satoru Wakasa

Department of Cardiovascular Surgery, Faculty of Medicine and Graduate School of Medicine, Hokkaido University, Sapporo, Japan

◐ These authors contributed equally to this work.
¤ Current address: Department of Cardiovascular Surgery, Carres Memorial Hospital, Sapporo, Japan
* shingu@huhp.hokudai.ac.jp

## Abstract

### Background

Inflammatory response plays a pivotal role in myocardial injury and post-infarction remodeling after acute myocardial infarction (AMI). Mechanical unloading (UL) of the left ventricle (LV) has been proposed as a potential therapeutic strategy to preserve cardiac function; however, its effects on myocardial inflammation remain incompletely understood.

### Methods

We employed a rat model of partial UL using heterotopic heart-lung transplantation following AMI. RNA sequencing (RNA-seq) was performed to evaluate transcriptomic changes, with a specific focus on inflammatory pathways in the non-infarcted remote area. Immune cell abundance was estimated using deconvolution analysis (QUANTISEQ). Quantitative PCR was performed to analyze some inflammatory cytokines, and macrophage polarization was evaluated by immunohistochemistry.

### Results

AMI significantly impaired cardiac function, which was mitigated by UL. RNA-seq analysis revealed marked activation of inflammatory pathways and identified several hub genes involved in cytokine signaling following AMI, while these transcriptional changes were not significantly altered in UL groups after AMI. Immune cell profiling demonstrated an increase in M2 macrophages after AMI, while UL preserved M2 macrophage levels. Histological analysis further supported UL's modulatory effect on macrophage polarization. Pro-inflammatory cytokines TNFα and IL1β were upregulated after AMI but showed attenuation with UL.

**Data availability statement:** All RNA-seq files are available from the ArrayExpress database (accession number(s) E-MTAB-16419).

**Funding:** This study was partly supported by JSPS KAKENHI Grant Number 22K08909 and JST SPRING, Grant Number JPMJSP2119. The funders had no role in study design, data collection and analysis, decision to publish, or preparation of the manuscript. There was no additional external funding received for this study.

**Competing interests:** The authors have declared that no competing interests exist.

## Conclusion

Partial UL potentially attenuates cardiac functional deterioration after AMI while exerting substantial effects on inflammatory gene expression and macrophage polarization. These findings suggest that the cardioprotective effects of UL may be correlated with the modulation of inflammatory pathways in the remote area after AMI.

## Introduction

Acute myocardial infarction (AMI) remains a leading cause of heart failure and death worldwide, despite substantial advances in reperfusion strategies and pharmacological therapies [1,2]. Myocardial ischemia initiates a robust inflammatory response that is essential for clearing necrotic debris and initiating tissue repair [3]. However, dysregulated or persistent inflammation may exacerbate myocardial injury and contribute to adverse remodeling and progressive heart failure [4].

Mechanical unloading (UL) of the left ventricle (LV), typically achieved through ventricular assist devices, has emerged as a promising therapeutic strategy to reduce myocardial wall stress, oxygen consumption, and infarct expansion following AMI [5], and has been increasingly applied in clinical settings, such as the percutaneous Impella device [6,7]. Several studies have demonstrated that UL can reduce infarct size, preserve cardiac function, and modulate myocardial metabolic stress when initiated during ischemia or at the time of reperfusion [8–10], while its clinical benefit needs further investigation. Furthermore, most existing studies have primarily focused on acute hemodynamic and metabolic effects, while the influence of prolonged UL, including immune responses, inflammatory signaling, and immune cell dynamics, remains poorly understood.

To address these issues, we employed an experimental model of a 2-week partial UL via heterotopic heart-lung transplantation in rats with a focus on viable myocardium in the remote area. Comprehensive transcriptomic profiling by RNA sequencing (RNA-seq) was performed. The results suggest that the molecular pathways of post-infarction inflammation were activated after AMI, which were attenuated by UL. Furthermore, immune cell abundance, inflammatory cytokine expression, and macrophage polarization were suppressed by UL. This integrated approach may provide novel insights into the interaction between mechanical UL and myocardial inflammation and offer potential therapeutic implications for improving cardiac function following AMI.

## Materials and methods

### Study protocol and animal models

Nine-week-old male Lewis rats were purchased from Japan SLC, Inc (Shizuoka, Japan). All animal experiments were conducted in accordance with the Hokkaido University Manual for Implementing Animal Experimentation and in compliance with the Guide for the Care and Use of Laboratory Animals of the US National Institutes of Health (NIH; publication No. 85-23, revised 1996). The study was approved by the Institutional Animal Care and Use Committee (No. 22-0126).

As shown in **Fig 1A**, the study included four groups: non-AMI (n = 6), AMI (n = 6), non-AMI/UL (n = 5), and AMI/UL (n = 5). AMI was induced by ligating the left anterior descending artery (LAD) using 7−0 polypropylene sutures (Ethicon; Raritan, NJ, USA). In non-AMI groups, the same procedures were performed without LAD ligation. According to our previous report, UL was performed after AMI via heterotopic heart-lung transplantation, which provided partial UL of the LV [11]. Briefly, heparin (1000 U; Mochida Pharmaceutical; Tokyo, Japan) was injected into the donor rat via the inferior vena cava. Cardiac arrest was induced by administration of 50-mL St. Thomas II solution. Finally, the donor heart and lung were excised and the ascending aorta of the donor heart was sutured to the abdominal aorta of the recipient rat (**Fig 1B**) within 1 h after AMI. General anesthesia was induced for all procedures by a single intramuscular injection of ketamine (90 mg/kg; Ketalar; Daiichi Sankyo Pharmaceutical, Tokyo, Japan) and xylazine (10 mg/kg; Selactar; Bayer Yakuhin; Osaka, Japan) followed by intubation and mechanical ventilation. After 14 days of UL, the rats were euthanized with intraperitoneal injection of secobarbital sodium (150 mg/kg; Aional; Nichi-Iko Pharmaceutical; Toyama, Japan). In the UL groups, only the donor hearts were excised for Langendorff perfusion and used for further experiments. For the first 5 days post-surgery, water containing aspirin (Fujifilm Wako Pure Chemical Corporation, Osaka, Japan) for pain relief (500 mg aspirin + 500 ml water) was provided as free-access drinking water. Thereafter, regular water was provided. Heterotopic heart transplantation was performed in syngeneic Lewis rats to prevent immunologic incompatibility without immunosuppression. Animals were administered an overdose of intraperitoneal pentobarbital (150 mg/kg) when they reached the humane endpoints, including difficulty eating or drinking, and signs of distress (self-mutilation, abnormal posture, respiratory distress, vocalization, etc.).

A

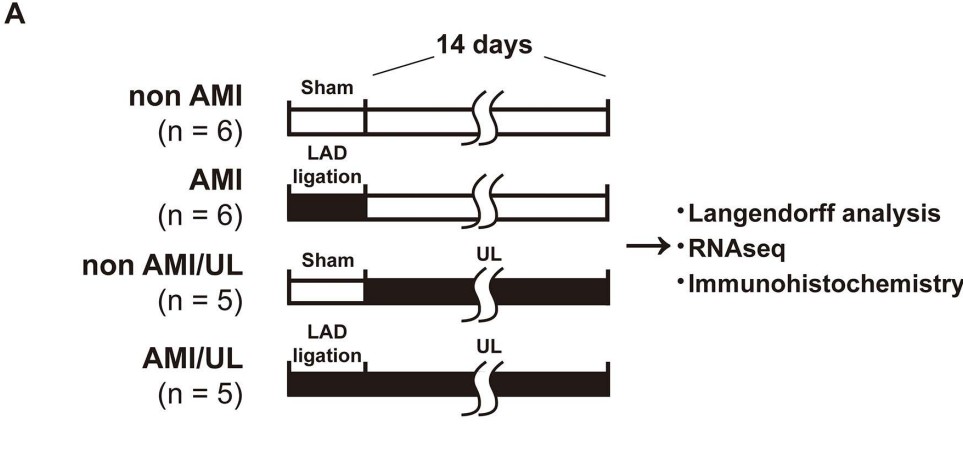

B

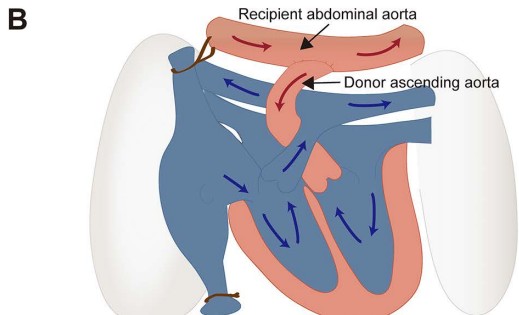

**Fig 1. Study protocol (A) and left ventricular UL model (B).** White bars denote periods without intervention; black bars represent periods with intervention (LAD ligation or unloading). AMI, acute myocardial infarction; LAD, left anterior descending artery; UL, unloading.

## Heart function analysis

Heart function was evaluated using echocardiography and a Langendorff perfusion system. Echocardiography was performed transthoracically or transabdominally using a Sonos 5500 ultrasound system with a 12-MHz phased-array transducer (Philips Medical Systems; Best, Netherlands). Under anesthesia by a single intramuscular injection of ketamine (90 mg/kg) and xylazine (10 mg/kg), heart rate (HR), LV end-diastolic dimension (LVEDD), and LV end-systolic dimension (LVESD) were measured from M-mode tracings obtained from the long-axis view of the LV. Echocardiography was conducted before the operation (baseline) and before euthanasia.

To assess cardiac function under constant loading conditions, the Langendorff perfusion was employed. The hearts were perfused with Krebs-Henseleit buffer. Its composition is provided in **Table S1** in S2 File. The buffer was oxygenated with 95% $O_2$ and 5% $CO_2$. The perfusion pressure was maintained at 60 mmHg, and the chamber temperature was set to 37°C. Upon stabilization, a 0.06-mL latex balloon was placed in the LV through the left atrium to measure LV pressure, which was recorded with PowerLab (ADInstruments; Dunedin, New Zealand) and analyzed using Lab-Chart (ADInstruments). The end-diastolic pressure of the LV was maintained at 5–10 mmHg. HR, coronary flow, LV developed pressure (LVDP), maximal rates of LV pressure rise and fall (maximum and minimum dP/dt), and rate pressure product (RPP = HR × LVDP) were acquired at the end of a 30-min perfusion.

The infarct area was identified as the macroscopically white region. The border area was defined as the area extending 3 mm from the infarct area to the free wall. The remaining zone was a remote area. The tissue samples were preserved in −80 °C for further investigation.

## Histology analysis

Fixed LV tissue blocks were sectioned and stained with hematoxylin–eosin to assess myocyte morphology. Cardiomyocyte cross-sectional area was quantified in non-infarcted myocardium by measuring 100 randomly selected myocytes containing a visible central nucleus per animal. Areas were outlined and calculated using ImageJ (NIH, 1.54f, U.S. National Institutes of Health; Bethesda, MD, USA).

Masson's trichrome staining was employed to determine the extent of fibrosis. For each animal, ten random microscopic fields from the non-infarcted zone were imaged, and collagen-rich regions were segmented using threshold-based detection in ImageJ 1.54f. Fibrosis was expressed as the percentage of stained area relative to total tissue area, and mean values per animal were used in group-level analyses.

## RNA sequence analysis

After the Langendorff perfusion, LV myocardial samples of the non-infarcted area were collected and stored at –80 °C. Total RNA was extracted using High Pure RNA Tissue Kit (Roche, Basel, Switzerland). RNA-seq was performed by Takara Bio Inc. (Kanagawa, Japan), and the data were uploaded into Annotare 2.0 with an accession number of E-MTAB-16419. Data analysis was conducted using the online application RNAseqChef (https://imeg-ku.shinyapps.io/RNA-seqChef_mirror2/; accessed Aug 18, 2025). Principal component analysis (PCA) and clustering were performed. Differentially expressed genes (DEGs) were identified with a false discovery rate < 0.05. K-means analysis was used to cluster DEGs into distinct expression pattern groups, k = 2, according to the Silhouette method, and the 2000 most significant genes were selected using a fold change > 1.5 between non-AMI and AMI groups. Multi-group DEG analysis was also performed on the same application using the same cut-off. Then, we focused on the genes that were markedly upregulated after AMI and maintained by UL. We performed pathway enrichment analysis using the Kyoto Encyclopedia of Genes and Genomes (KEGG) database and explored the connections between genes and pathways using cnetplot in RNAseqChef. The genes connected to multiple pathways in the cnetplot were regarded as hub genes.

## Immune cell infiltration

Because we identified inflammatory pathways by the aforementioned classification analysis, we deconvolved the immune cell fraction from the row data of RNA-seq using QUANTISEQ (http://icbi.at/quantiseq; accessed Aug 18, 2025). Although QUANTISEQ was originally developed for human tumor RNA-seq data, we applied it to rat myocardial RNA-seq because no rat-specific immune deconvolution pipelines are currently available and many immune-cell signature genes are evolutionarily conserved across species. We used the default human TIL10 signature matrix and performed deconvolution on normalized TPM data from our bulk RNA-seq. Subsequently, we performed a clustering analysis of immune cell numbers across the four groups and analyzed the association between the hub gene expression and cell numbers using R version 4.3.2 (R Core Team, Vienna, Austria).

## Inflammatory cytokine analysis

To validate the RNA-seq results, the expression levels of inflammatory cytokines in LV myocardial samples from the remote areas were analyzed. mRNA levels of interleukin (IL) 1β, IL6, IL10, and tumor necrosis factor (TNF) α were quantified by RT-qPCR. Total RNA was reverse transcribed into cDNA using the Transcriptor First Strand cDNA Synthesis Kit (Roche). Quantitative PCR was performed using the FastStart Essential DNA Probes Master (Roche). The sequences of the primers and probes were listed in **Table S2** in S2 File. The relative gene expressions were analyzed using a housekeeping gene Rsp29.

## Immunohistochemistry

Based on the results from QUANTISEQ, we focused on macrophages. We performed immunofluorescence staining using LV tissues. Briefly, LV myocardial samples were fixed in 3.5% neutral buffered formalin and embedded in paraffin. After deparaffinization and rehydration, antigen retrieval was performed using Dako Target Retrieval Solution, pH 9.0 (Dako S2367, Singapore), at 95°C for 20 minutes. Sections were incubated overnight at 4 °C with rabbit recombinant multiclonal anti-iNOS antibody (1:500 dilution, ab283655, Abcam, Cambridge, UK), anti-CD163 antibody (1:500 dilution, ab316218, Abcam), and goat polyclonal anti-SPP1 antibody (1:100 dilution, ab11503, Abcam) for the detection of M1 macrophages, M2 macrophages, and SPP1+ macrophages, respectively. Following primary antibody incubation, slides were washed and incubated for 30 minutes at room temperature with the corresponding secondary antibodies: Goat anti-Rabbit IgG (H+L) Highly Cross-Adsorbed Secondary Antibody, Alexa Fluor™ Plus 594 (1:100 dilution, A32740, Invitrogen, CA, USA) and fluorescent (FITC)-conjugated AffiniPure Goat Anti-Mouse IgG (H+L) antibody (1:100, AB_2338589, Jackson ImmunoResearch Laboratories, Inc, Ely, UK). Nuclei were counterstained with 4′,6-diamidino-2-phenylindole (DAPI). Stained sections were imaged using a fluorescence microscope (BZ-X, Keyence Corporation, Osaka, Japan). For quantitative analysis, positive cells were detected in randomly selected fields using Image J and the ratio of positive-cell area to field area was calculated.

## Statistical analysis

All data in this study are presented as mean ± SEM. Normality was evaluated using residual-based diagnostics, with visual inspection of Q–Q plots (**Fig S1** in S1 File). Because UL is implemented via a transplantation-based model, it inevitably introduces biological effects independent of AMI and therefore cannot be regarded as a neutral sham condition. To address this limitation, we adopted a four-group design analogous to knockout (KO) studies (non-AMI, AMI, non-AMI/UL, and AMI/UL) and analyzed the data using two-way ANOVA with AMI and UL as factors. The interaction term (AMI × UL) was used to assess whether UL influenced the time course of AMI. Post hoc Bonferroni tests were conducted between non-AMI and AMI, and between non-AMI/UL and AMI/UL. Effect sizes (partial η²) were calculated using Python 3.11.12 (Python Software Foundation, Wilmington, DE, USA) in Google Colaboratory (Google Research, Mountain View, CA, USA, https://colab.research.google.com/; accessed Apr 30, 2025) for parameters with $p < 0.2$. Some functional and histological data were collected from the same animal cohort as our previous publication [11]. Statistical analysis was

performed with R 4.3.0 and Prism 9.5.1 (GraphPad, San Diego, CA, USA). A two-tailed $p < 0.05$ was considered statistically significant.

## Results

### Heart function and pathological examinations

The body weight, heart weight, and echocardiographic parameters are presented in **Tables S3 and S4** in S2 File. There were no differences in the baseline values. The heart size was significantly reduced by 14-day UL.

Because echocardiographic assessment under varying loading conditions is not appropriate, cardiac function was evaluated using isolated heart perfusion to assess the effects of AMI and UL (**Fig 2A**). HR and coronary flow were comparable among groups. AMI markedly reduced cardiac contractility and relaxation, as reflected by decreases in minimum dP/dt and RPP. Although unloading did not produce statistically significant improvements in functional parameters, the AMI/UL group demonstrated a modest trend toward attenuation of AMI-induced decline, with RPP values approaching those of the non-AMI/UL group. The effect size for the interaction (partial $\eta^2 = 0.085$; threshold for moderate effect $= 0.06$) suggests a potential influence of unloading on functional adaptation, despite the absence of a significant AMI × UL interaction. These observations indicate that the degree of partial unloading applied in this model may influence post-infarction function, although the magnitude of its effect on global pump function was limited.

In pathological examinations (**Fig 2B**), there was no interaction (AMI × UL) in either parameter (myocyte area or fibrosis), and partial $\eta^2$ values were small ($< 0.06$). This suggests that UL did not influence the AMI time course in terms of myocyte remodeling and fibrosis. Also, there were no significant interaction of myocyte area and fibrosis in the border zone (**Fig S2** in S1 File).

### RNA-seq analysis and hub gene identification

As shown in **Fig 3A**, the AMI group was separated from the non-AMI group, whereas no clear separation was observed between the AMI/UL and non-AMI/UL groups in the PCA plot. Compared to the DEGs between the non-AMI and AMI group (**Fig 3B**), there were fewer DEGs between the UL groups (**Fig 3C**). AMI changed gene expression in both groups with or without UL, with more genes upregulated than downregulated. The results of other group comparisons are shown in **Fig S3** in S1 File.

We used DEGs with a $p$ value $< 0.05$ and clustered them by RNAseqChef according to their expression among four conditions. The results of k-means analysis are shown in **Fig 4A**. Genes in cluster 1 were related to fatty acid metabolism, oxidative phosphorylation, adipogenesis, peroxisome, and bile acid metabolism (**Fig 4B**). The expression of *Cpt1b, CD36l1, and Fabp3*, which are related to fatty acid metabolism, was significantly decreased after AMI (**Fig 4C**). In cluster 2, the genes were mainly related to epithelial–mesenchymal transition, TNFα signaling via NF-κB, allograft rejection, inflammatory response, and IL-6–JAK–STAT3 signaling (**Fig 4B**). Because cluster 2 contained the largest number of genes, inflammation may represent the most prominent difference among the four groups.

Multi-group DEG analysis was also performed for different expression patterns among four groups (**Fig 5A**). There were 10 groups identified. In groups 1 and 9, there was a marked increase in gene expression between the non-AMI and AMI groups, while almost no difference between the non-AMI/UL and AMI/UL groups. Although groups 8 and 10 also showed different trends in gene expression depending on the presence or absence of UL following AMI, the number of affected genes was too small to allow for meaningful further analysis. Because we aimed to find the genes that were most related to UL, we focused on groups 1 and 9 in the next analysis.

**Fig 5B** shows the result of pathway enrichment analysis. Group 1 was associated with cytokine-cytokine receptor interaction, graft versus host disease, and intestinal immune network. Group 9 included natural killer cell mediated cytotoxicity, T cell and B cell receptor pathways. In both groups 1 and 9, pathways related to inflammation and immune cells were highly enriched. Quantitative analysis of related hub genes revealed that the expression levels were significantly

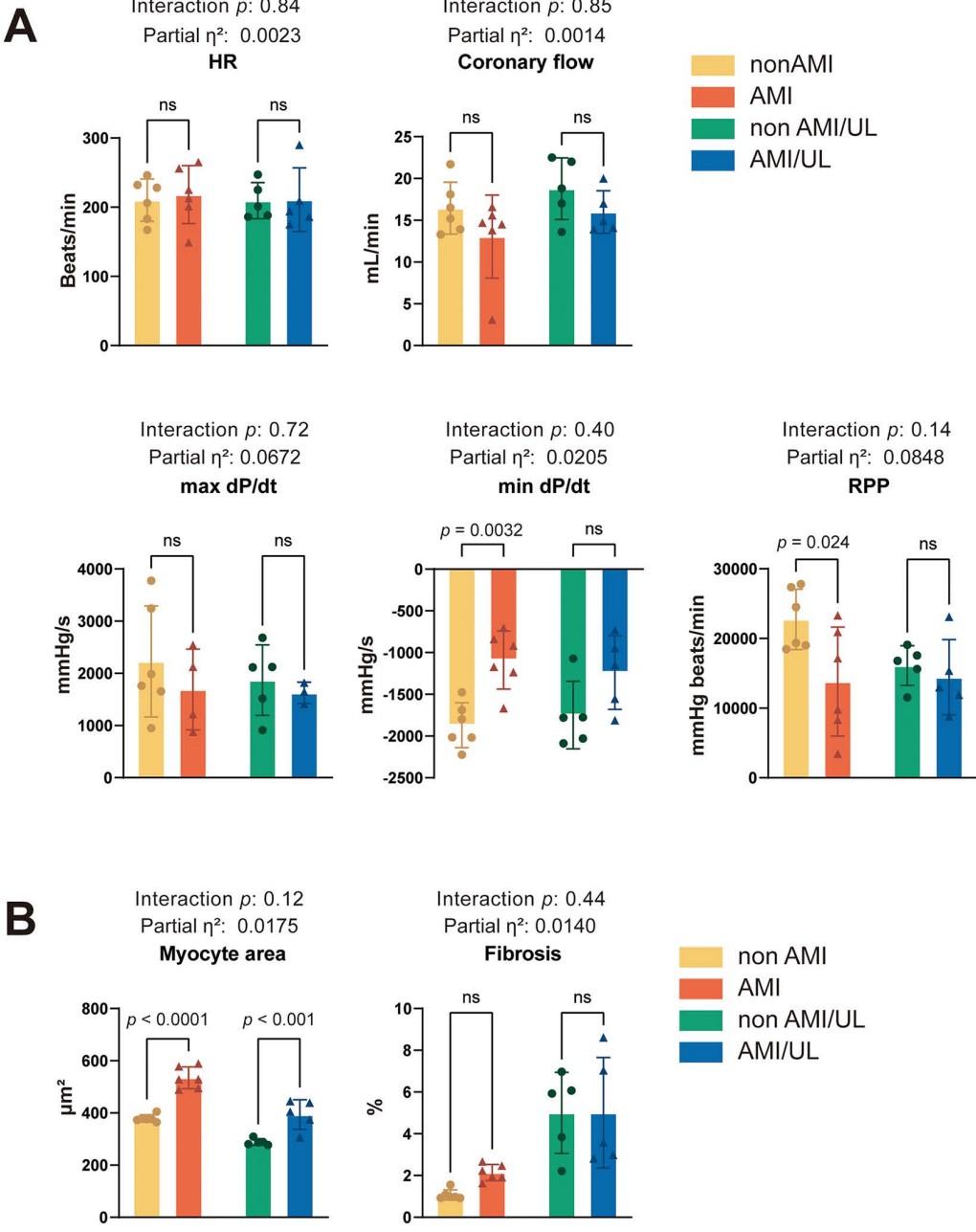

**Fig 2. Cardiac functional parameters (A) and myocyte size and fibrosis (B) 14 days after surgery.** Data are presented as mean ± SEM. Statistical significance was assessed using two-way ANOVA, followed by Bonferroni post hoc test. Interaction p-values and effect sizes (partial η² values) are shown. *P < 0.05, **P < 0.01, ***P < 0.001, ****P < 0.0001. AMI, acute myocardial infarction; HR, heart rate; ns, not significant; RPP, rate pressure product; UL, unloading.

increased after AMI and remained unchanged following UL in most genes (**Fig 5C**). The significant interactions (AMI × UL) were observed in Interleukin-1 receptor type 2 (IL1r2), Interleukin-2 receptor alpha chain (IL2ra), and Rat MHC class II molecule (RT1 Db1) in group 1. Although the *p* values were not significant, the effect sizes of interaction were large in Lymphocyte cytosolic protein 2/ SLP-76 (Lcp2) and Protein kinase C beta (Prkcb) in group 9.

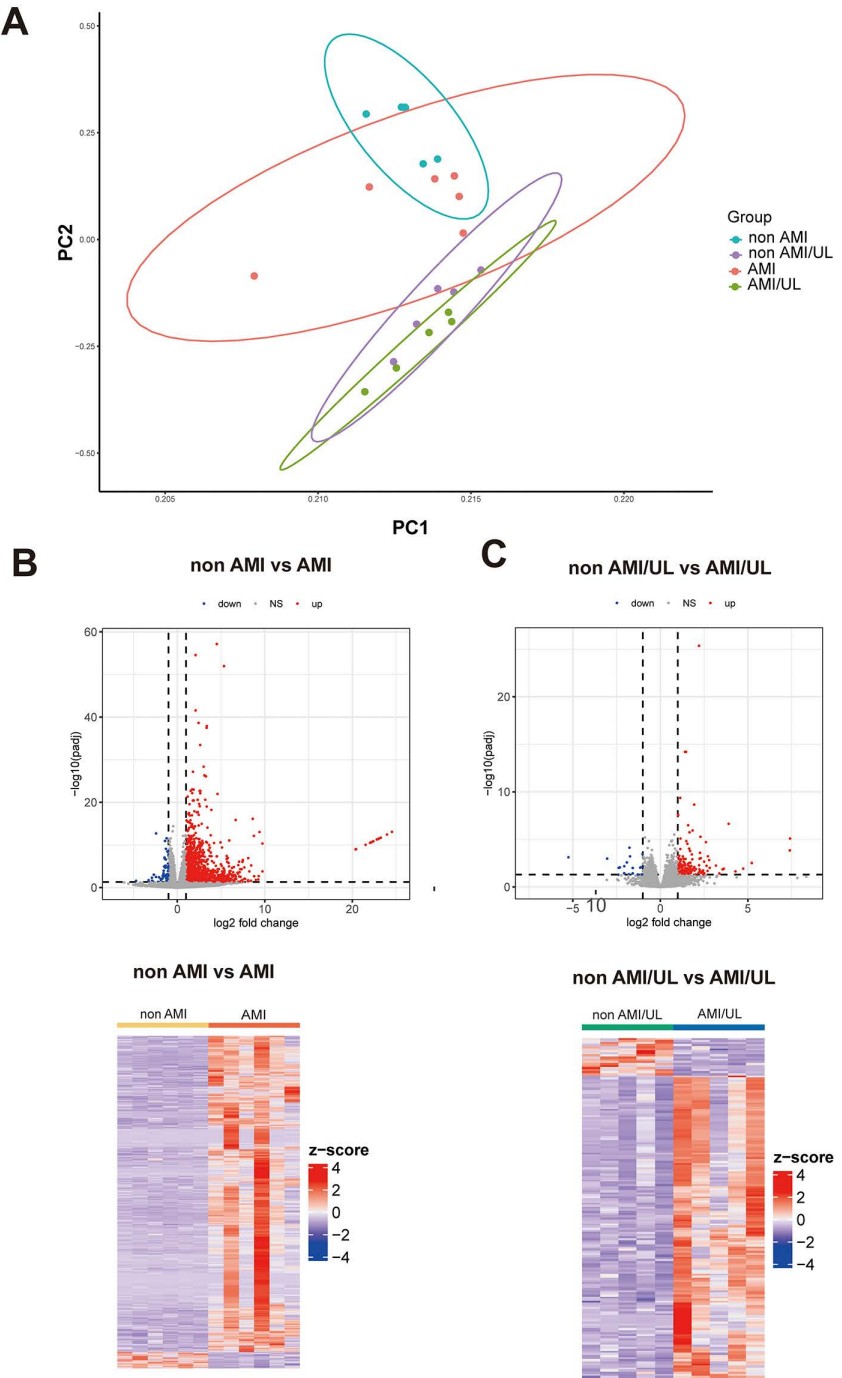

**Fig 3. RNA-seq analysis of the LV myocardium in the non-infarcted area 14 days after surgery. (A)** Principal component analysis (PCA) of RNA-seq data (rlog-transformed counts). PC1 and PC2 explain 35.2% and 17.8% of total variance, respectively. Samples are color-coded by treatment group (non-AMI: light Blue; AMI; red; non-AMI with UL: purple; and AMI with UL: green). Each point represents an individual biological replicate (n = 5 or 6 per group). **(B and C)** Volcano plot of differentially expressed genes between non-AMI and AMI, non-AMI with UL and AMI with UL. Genes with adjusted p < 0.05 and |log$_2$ fold change| > 1.5 are considered significantly differentially expressed (red: upregulated; blue: downregulated; gray: not significant). Hierarchical clustering heatmap indicates the differentially expressed genes. Rows represent genes and columns represent individual samples. Color scale indicates relative expression levels (red: high; blue: low). AMI, acute myocardial infarction; LV, left ventricle; UL, unloading.

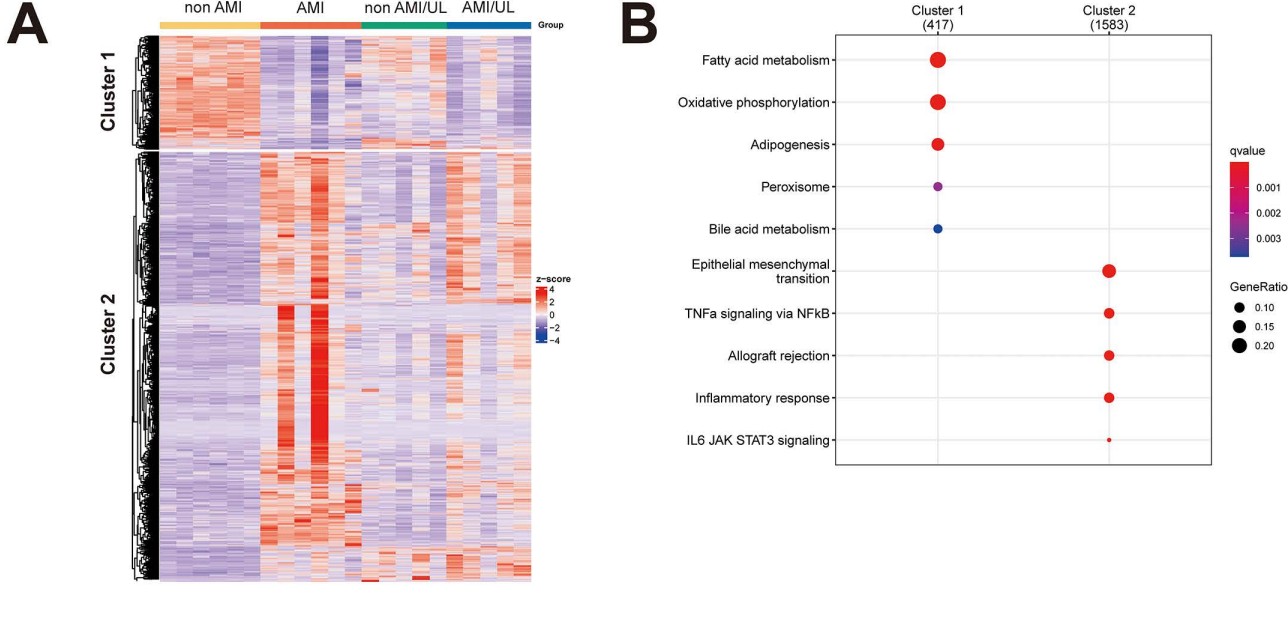

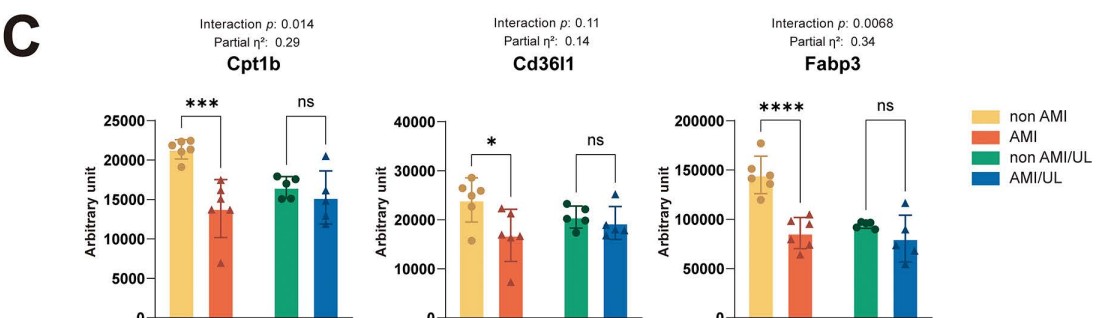

**Fig 4. Gene clustering from k-means analysis. (A)** Heatmap of two gene expression clusters across the four groups (non-AMI, AMI, non-AMI/UL, and AMI/UL). **(B)** KEGG pathway enrichment for each cluster showing metabolism-related pathways in Cluster 1 and inflammation/immune-related pathways in Cluster 2. K-means analysis was used to cluster DEGs into distinct expression pattern groups, k = 2, according to the Silhouette method, and the 2000 most significant genes were selected using a fold change > 1.5 between non-AMI and AMI groups. **(C)** Gene expressions for *Cpt1b*, *Cd36l1*, and *Fabp3*. Data are presented as mean ± SEM. Statistical significance was assessed using two-way ANOVA, followed by Bonferroni post hoc test. Interaction p-values and effect sizes (partial η² values) are shown. *P < 0.05, ***P < 0.001, ****P < 0.0001. AMI, acute myocardial infarction; LV, left ventricle; UL, unloading.

## Immune cell infiltration and its association with hub genes

Immune cell fraction in LV myocardial tissues was predicted using QUANTISEQ across the four experimental groups. As shown in **Fig 6A**, the proportion of M2 macrophages was significantly increased after AMI compared to non-AMI (*p* < 0.001), indicating activation of an anti-inflammatory response 14 days after myocardial injury. On the other hand, UL preserved M2 macrophage levels, as no significant differences were observed between non-AMI/UL and AMI/UL groups. Furthermore, the interaction (AMI × UL) had a moderated effect size (0.06). Other immune cell subsets, including monocytes, NK cells, and T cells (CD4+, CD8+, regulatory T cell), showed no significant differences among groups. Consistent with these findings, hierarchical clustering heatmap further demonstrated distinct patterns of immune cell composition, with increased M2 macrophage infiltration predominantly in AMI groups (**Fig 6B**). Furthermore, correlation analysis of hub

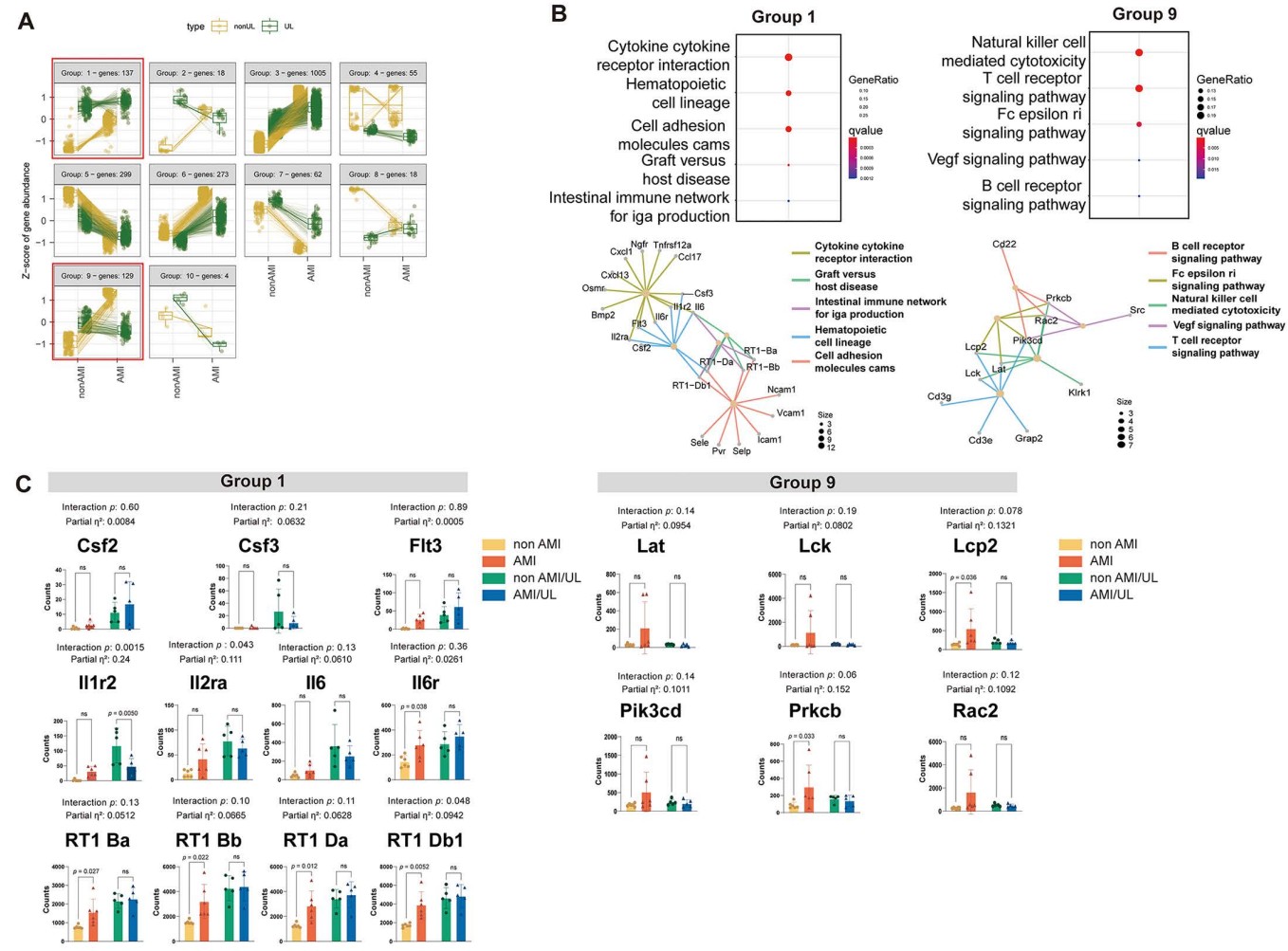

**Fig 5. Hub gene detection related to inflammation from the RNA-seq analysis. (A)** Classification of DEGs into distinct expression pattern groups using a fold change > 1.5 between the non-AMI and AMI groups as a cut-off. The genes in groups 1 and 9 were greatly elevated after AMI, and the elevation was suppressed by UL. **(B)** Pathway analysis using KEGG database and protein-protein interaction network to find hub genes from shared genes. **(C)** Comparison of the hub genes among the four groups.. Data are presented as mean ± SEM. Statistical significance was assessed using two-way ANOVA, followed by Bonferroni post hoc test. Interaction p-values and effect sizes (partial η² values) are shown. *P < 0.05, **P < 0.01. AMI, acute myocardial infarction; DEG, differentially expressed genes; KEGG, Kyoto Encyclopedia of Genes and Genomes; UL, unloading.

gene expression with immune cell abundance revealed that hub genes from RNA-seq were largely positively correlated with immune cell activation in AMI (**Fig 6C**).

## Immune cytokine expressions

The expression of inflammatory cytokines in the LV non-infarct areas was assessed by RT-qPCR (**Fig 7**). TNFα and IL1β levels were significantly elevated in the AMI group compared to the non-AMI group, indicating enhanced pro-inflammatory response following AMI. UL significantly attenuated the expression of both TNFα and IL1β. In contrast, IL10, an anti-inflammatory cytokine, was also significantly increased in the AMI group compared to the non-AMI group, suggesting a compensatory anti-inflammatory response. UL suppressed this AMI-induced IL10 upregulation.

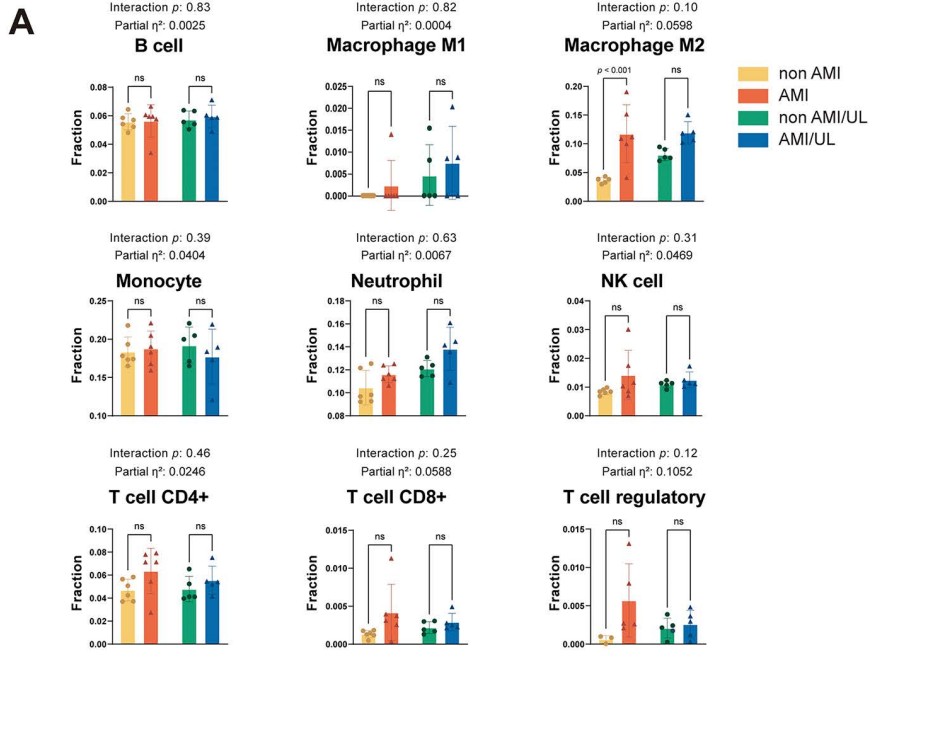

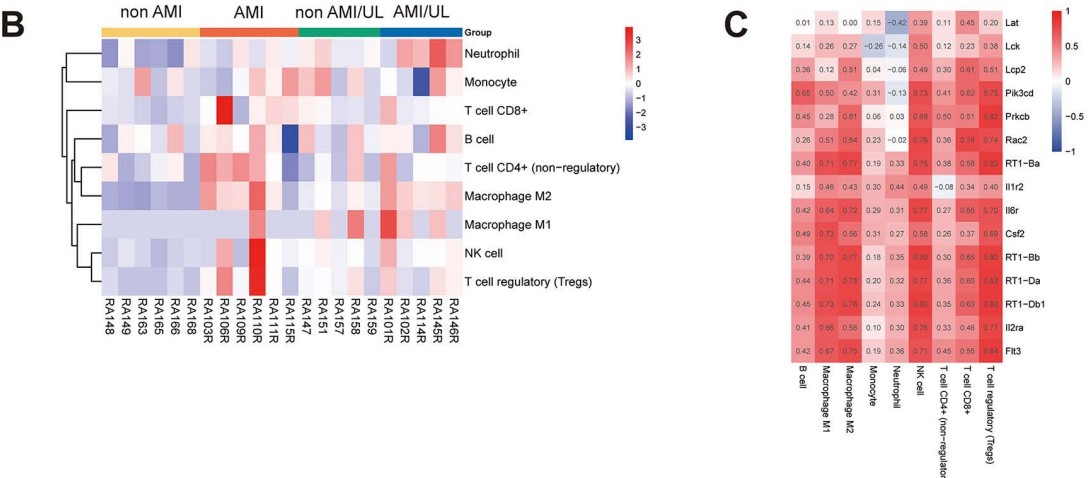

**Fig 6. Deconvolved immune cell type expression from the raw RNA-seq data using QUANTISEQ. (A)** Comparison of the cell type fractions among the four groups. **(B)** Heatmap of the correlation between immune cell types and each rat of the four groups. **(C)** Association between the hub gene expressions and cell types. Data are presented as mean ± SEM. Statistical significance was assessed using two-way ANOVA, followed by Bonferroni post hoc test. Interaction p-values and effect sizes (partial η² values) are shown. ***P < 0.001. AMI, acute myocardial infarction; UL, unloading.

## Immunohistochemistry for macrophages

To further validate the alterations in macrophage polarization, immunohistochemical analysis of M1 (iNOS⁺) and M2 (CD163⁺) macrophages was performed in non-infarcted areas (**Fig 8A**). Consistent with the previous immune cell composition and gene expression results, M2 macrophage levels were significantly increased after AMI compared to non-AMI,

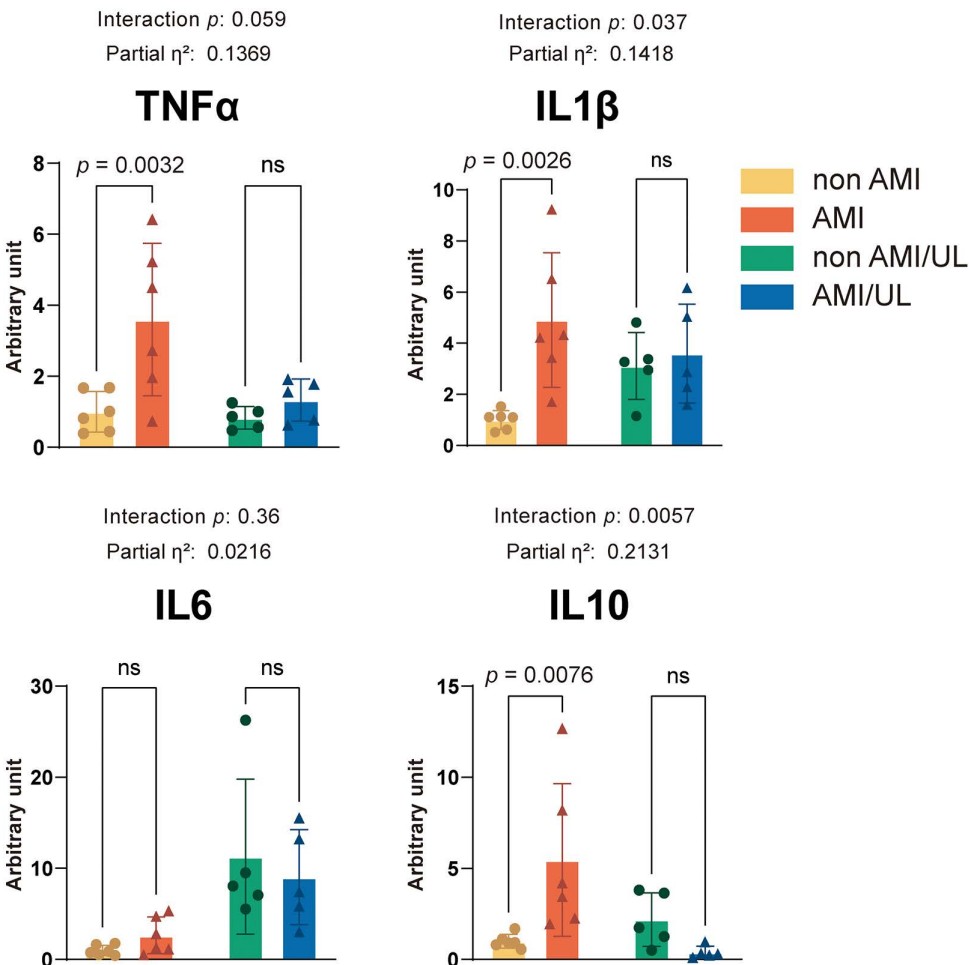

**Fig 7. Gene expression levels of inflammatory cytokines in the LV myocardial samples of remote areas 14 days after surgery.** Data are presented as mean±SEM. Statistical significance was assessed using two-way ANOVA, followed by Bonferroni post hoc test. Interaction p-values and effect sizes (partial η² values) are shown. **P<0.01. AMI, acute myocardial infarction; IL, interleukin; LV, left ventricle; and TNF, tumor necrosis factor; UL, unloading.

while UL attenuated the increase in M2 macrophages after AMI (interaction $p = 0.0082$; partial $\eta^2 = 0.30$; threshold for large effect = 0.14). M1 macrophages did not differ significantly among groups. Additionally, macrophage distribution was evaluated in both the infarct and border zones (**Fig S4A** in S1 File). In the infarct area, AMI led to significant increases in both M1 and M2 macrophage infiltration, and no AMI×UL interaction was observed. In the border zones, M2 macrophages were also elevated in AMI compared to non-AMI, whereas M1 macrophage levels and M1/M2 ratios remained unchanged across groups and there was no interaction between AMI and UL.

Given that pro-inflammatory cytokines (TNFα and IL1β) are primarily secreted by M1 macrophages, we analyzed SPP1+ M2 macrophages using immunostaining, because a subset of SPP1 + M2 macrophages can also produce inflammatory cytokines [6,12]. **Fig 8B** shows the immunostaining of iNOS, CD163 and SPP1 in the non-infarcted region. While double-positive macrophages were observed in the infarcted area and border zone (**Fig S4B** in S1 File), no such cells were detected in the non-infarcted region. This suggests that the SPP1+ M2 macrophages are unlikely to account for the increase in inflammatory cytokines, and the primary origin of the cytokines may originate from other cells.

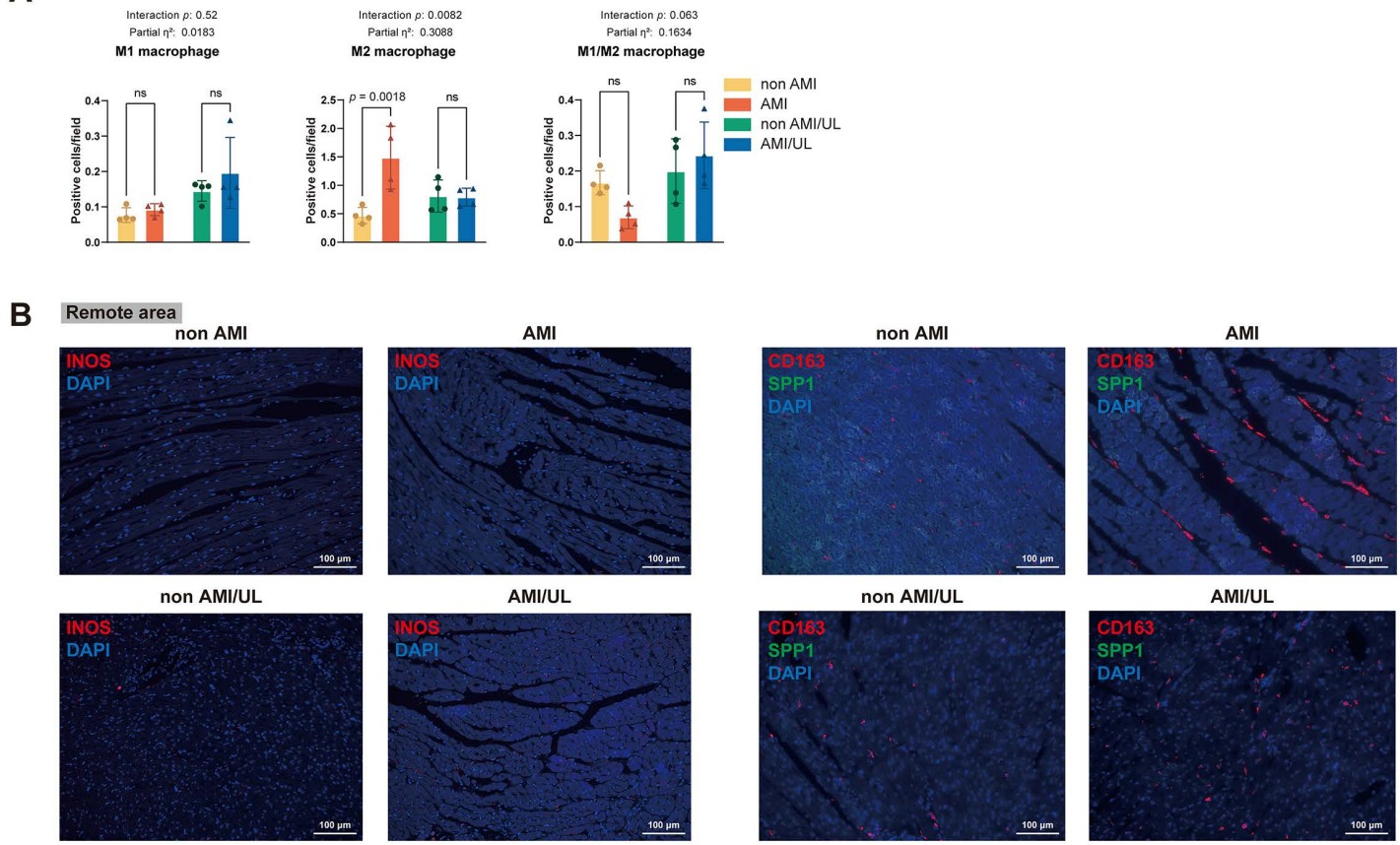

**Fig 8. M1, M2 and SPP1 positive macrophages from remote areas 14 days after surgery. (A)** Quantification of M1, M2, and M1/M2 macrophages across groups (non-AMI, AMI, non-AMI/UL, and AMI/UL). Data are shown as positive cell fractions with statistical comparisons indicated. **(B)** Representative immunofluorescence staining of iNOS and CD163 (red), SPP1 (green), and DAPI (blue) in the remote myocardium. No SPP1+ cells were detected in all groups. Scale bars, 100 µm. Data are presented as mean±SEM. Statistical significance was assessed using two-way ANOVA, followed by Bonferroni post hoc test. Interaction p-values and effect sizes (partial η² values) are shown. **P<0.01. AMI, acute myocardial infarction; UL, unloading.

## Discussion

The main findings of this study were that AMI induces sustained expression of inflammatory cytokines and marked accumulation of M2 macrophages two weeks after AMI, consistently observed across myocardial regions including the infarcted area, border zone, and non-infarcted region. In contrast, UL had limited influence on macrophage polarization, primarily restricted to the non-infarcted region. Moreover, given that the increased M2 macrophages were negative for SPP1 expression, other cell types may represent alternative sources of the inflammatory cytokines. Unlike prior studies focusing mainly on infarct or border zones, our analysis highlights the importance of the viable remote area as a key site of persistent immune modulation after AMI.

### Inflammation in heart failure induced by myocardial infarction

Numerous previous studies have focused on inflammatory activation within the infarct zone, where massive cardiomyocyte necrosis triggers robust immune cell infiltration and cytokine release [13]. In addition, circulating inflammatory cytokines, for example TNFα, IL1β, and IL6, have been widely evaluated as prognostic indicators in patients with AMI and

heart failure [14,15]. The accumulation of these inflammatory cells and pro-inflammatory cytokines contributes to tissue injury, extracellular matrix degradation, and adverse remodeling [16]. However, accumulating evidence suggests that inflammation may also affect the remote, non-infarcted myocardium, contributing to adverse remodeling and progressive dysfunction. A study using autopsy from AMI patients suggests that immune cell infiltration also increased in the unaffected viable remote area, even though it is less pronounced than in the infarct area [17]. Consistent with these findings, a pilot clinical study using MRI T2 mapping revealed that inflammatory changes in remote myocardium were associated with LV remodeling after AMI [18]. Furthermore, a preclinical study using the promising inflammatory imaging agent [⁶⁸Ga] Ga-Pentixa in an AMI rat model also demonstrated that inflammation increased in the remote myocardium and was strongly correlated with contemporaneous end-diastolic and systolic volumes [19].

In this study, we also examined inflammatory responses in the non-infarcted remote myocardium following AMI. RNA-seq analysis demonstrated marked upregulation of multiple inflammatory pathways and hub genes related to cytokine signaling and immune activation, indicating that inflammatory activation extends beyond the infarct core to the viable remote myocardium. Among these inflammation-related hub genes, *Il1r2*, *Il2ra*, and *RT1-Db1* exhibited significant AMI × UL interaction effects, suggesting that their expression is specifically modulated by UL in post infarction. Notably, previous clinical evidence has shown that circulating *Il1r2* levels in patients with myocardial infarction are independently associated with parameters of adverse left ventricular remodeling [20]. In our study, UL alone caused an increase in *Il1r2* expression, and it was minimally affected by AMI. Thus, the suppression of *Il1r2* in the AMI/UL group may possibly reflect a shift in inflammatory signaling toward a transcriptional profile associated with less adverse remodeling, although causality cannot be inferred from the present data. No studies have directly linked *Il2ra* or *RT1-Db1* to AMI or mechanical unloading. Therefore, our data provide novel evidence that ventricular UL modulates inflammation-related gene programs in the remote myocardium after AMI, highlighting a previously underexplored interaction between UL and post-AMI inflammatory regulation. These results were further validated by RT-qPCR. The anti-inflammatory cytokine IL10 was also upregulated in the AMI group, which may represent a compensatory feedback mechanism to counteract increased inflammatory signaling. These findings suggest that inflammation is not limited to the infarct area but may extend to non-infarcted regions, potentially contributing to global ventricular function and remodeling after AMI. Canonical Wnt/β-catenin signaling has been implicated in post-MI inflammation and fibrotic remodeling in previous studies [21,22]. However, in the present study, the expression of representative canonical Wnt markers (Ctnnb1, Axin2, Lrp6, and Tcf7l2, data not shown) was not significantly altered among groups, and no interaction between MI and unloading was detected. These data suggest that canonical Wnt activation was not prominent in the remote myocardium under our experimental conditions, and that the inflammatory modulation observed here may be governed by other signaling mechanisms.

## Macrophages in heart failure

Macrophage polarization represents a critical regulator of post-AMI repair and is differentially influenced by mechanical unloading. However, studies on macrophages during UL after AMI remain limited. Zhou et al. examined macrophage dynamics in both infarcted and non-infarcted myocardial regions using a heterotopic heart transplantation model (complete UL). They observed that UL delayed the resolution of inflammation in the infarcted area, as indicated by prolonged macrophage infiltration. However, in the non-infarcted myocardium, immune cell infiltration was transient and not sustained beyond the early postoperative phase, suggesting that UL itself does not promote persistent inflammation in the remote area [23]. In contrast, our results from QUANTISEQ suggested that AMI led to an increase in M2 macrophage counts, and it was further supported by immunohistochemical analysis. We observed sustained M2 macrophage accumulation in the remote myocardium 14 days post-AMI, which was significantly attenuated by partial UL. Furthermore, UL tended to selectively suppress M1 polarization (partial $\eta^2 = 0.0606$: moderate effect) in the infarct zone without significantly affecting M2 levels (partial $\eta^2 = 0.0011$: no effect). These findings suggest that mechanical UL exerts region- and phenotype-specific immunomodulatory effects, differing from prior reports and highlighting the importance of assessing

macrophage polarization rather than total macrophage counts. The discrepancy between the results of Zhou et al. and ours could be attributed to the difference in the models: complete vs. partial UL. A significant reduction in heart rate may have contributed to delayed lymphatic drainage of macrophages in the infarct zone in their complete UL model, while heart rate was not altered by UL in our model.

Macrophage polarization plays a critical role in post-AMI inflammation and repair. Pro-inflammatory M1 macrophages predominate in the early phase, whereas M2 macrophages contribute to inflammation resolution and tissue remodeling. A timely transition from M1 to M2 is essential for optimal healing, and disruption of this process has been linked to adverse remodeling. To our knowledge, this is the first study to report macrophage polarization during UL after AMI, showing that M2 macrophages increased following AMI but remained at similar levels among the UL groups. Regarding the potential mechanisms underlying the M1-to-M2 transition, recent evidence indicates that cellular metabolic status, particularly energy metabolism, is a critical regulator of macrophage polarization [24]. AMP-activated protein kinase (AMPK) is a key cellular energy sensor that has emerged as an important modulator of immune cell function, including macrophage polarization. Activation of AMPK promotes M2 polarization while suppressing pro-inflammatory M1 phenotypes. Mounier et al. demonstrated that AMPKα1 is required for the transition of macrophages from M1 to M2 during the resolution of inflammation, in which AMPKα1-deficient macrophages exhibit impaired acquisition of M2 markers and defective phagocytosis-driven phenotype switching [25]. Given that UL decreases myocardial metabolic demand and, as shown in our previous study [11], suppresses AMPK activation, UL is therefore likely to contribute to the reduced M2-dominant macrophage polarization in the remote area.

Because M2 macrophages increased only in the AMI group, we considered whether there might be other explanations for this result. Reggio et al. reported that SPP1+ macrophages often display M2-like features, while also expressing certain pro-inflammatory (M1-like) genes, such as TNFα and IL1β [12]. This could be one reason for the increased M2 macrophages in the AMI group. However, in our study, the increased M2 macrophages were negative for SPP1 expression in the AMI group, suggesting that the cardiomyocytes or other immune cell populations may represent the principal source of inflammatory cytokines.

### Limitations

This study has several limitations. First, although we observed that UL attenuated inflammatory cytokine expression and was associated with preserved LV function after AMI, a direct causal relationship between inflammation and LV functional changes could not be established. Future studies should investigate whether suppression of inflammation after AMI and UL directly influences LV function, and should elucidate the mechanistic link between immune modulation and cardiac performance. Second, other M2-type macrophage subsets capable of producing inflammatory cytokines were not examined. Alexian et al. reported that in an animal model of pressure overload, IL1β derived from Cx3Cr1-expressing macrophages modulated fibroblast states, and IL1β neutralization improved outcomes in heart failure [26]. Third, only male rats were used, which may limit generalizability, and whether sex modifies the effects of unloading after MI should be evaluated in the future. Finally, we did not quantify infarct size, cardiomyocyte apoptosis, and capillary and arteriolar density in this study. Because only a single papillary-level section was collected per heart, and specialized staining is necessary, reliable estimation was not feasible. Future studies will incorporate serial histological sectioning or whole-heart TTC staining to accurately determine infarct size and evaluate whether unloading affects acute myocardial injury.

### Conclusions

Partial UL suggests a potential influence on cardiac function after AMI and exerts effects on inflammatory gene expressions as well as macrophage polarization in the non-infarcted area. These findings suggest that mechanical UL may confer cardioprotective effects through modulation of localized inflammatory responses within the viable remote myocardium, at least in part. Nevertheless, the coexistence of fibrosis and myocardial atrophy highlights the complex interplay between

the beneficial effects of UL and maladaptive remodeling. A deeper understanding of the molecular mechanisms underlying this balance will be critical for optimizing UL strategies to improve therapeutic outcomes in ischemic heart failure.

## Supporting information

**S1 File. Normality test for the parameters, cardiac myocyte size and fibrosis in the border zone 14 days after surgery, differential gene expression analysis between groups, and macrophage polarization in infarct areas and border zones.**
(DOCX)

**S2 File. Composition of the Krebs-Henseleit buffer, sequences of the primers and probes, body weight, LV weight, LV/body weight after 14 days, and echocardiographic parameters.**
(DOCX)

## Acknowledgments

We appreciate the technical support provided by Keyence Corporation for the use of fluorescence microscopy and Sapporo General Pathology Laboratory Co., Ltd. for their valuable technical assistance related to pathology. We acknowledge the assistance of ChatGPT (OpenAI) for language editing and improvement of the manuscript. The authors are fully responsible for the scientific content and interpretation.

## Author contributions

**Conceptualization:** Yasushige Shingu, Satoru Wakasa.

**Data curation:** Jingwen Gao, Yasushige Shingu, Ryota Azuma.

**Formal analysis:** Jingwen Gao, Yasushige Shingu.

**Funding acquisition:** Yasushige Shingu.

**Investigation:** Jingwen Gao, Yasushige Shingu.

**Methodology:** Jingwen Gao, Yasushige Shingu, Ryota Azuma.

**Software:** Jingwen Gao.

**Supervision:** Yasushige Shingu, Satoru Wakasa.

**Validation:** Jingwen Gao, Yasushige Shingu.

**Visualization:** Jingwen Gao.

**Writing – original draft:** Jingwen Gao, Yasushige Shingu.

**Writing – review & editing:** Jingwen Gao, Yasushige Shingu, Ryota Azuma, Satoru Wakasa.

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
