## [Decision Letter · Decision Letter 0]

1 Dec 2025

Dear Dr. Shingu,

Thank you for submitting your manuscript to PLOS ONE. After careful consideration, we feel that it has merit but does not fully meet PLOS ONE’s publication criteria as it currently stands. Therefore, we invite you to submit a revised version of the manuscript that addresses the points raised during the review process.

**ACADEMIC EDITOR:** The current form requires a careful point-by-point revision and additional data are expected.

We look forward to receiving your revised manuscript.

Kind regards,

Vincenzo Lionetti, M.D., PhD

Academic Editor

PLOS ONE

Journal Requirements:

“This study was partly supported by JSPS KAKENHI Grant Number 22K08909.”

“This study was partly supported by JSPS KAKENHI Grant Number 22K08909.”

**Additional Editor Comments:**

Major issues

1) The manuscript provides valuable insights into post-infarction remodeling and the effects of LV unloading but fails to consider the potential involvement of the canonical Wnt/β-catenin pathway, a key regulator of inflammation and fibrosis after MI (please see Cardiovasc Res, 2016;112:645–655). Given the overlap between the authors’ findings and Wnt-regulated processes, this omission limits the mechanistic depth of the study. The authors should analyze canonical Wnt pathway activation using their existing RNA-seq dataset (i.e.: KEGG/GSEA enrichment or marker gene expression such as CTNNB1, AXIN2, LRP6, TCF7L2) to strengthen the biological interpretation. Finally, they should discuss the results in the light of abovementioned study.

2) The authors should complete the histological characterization of hallmarks of myocardial remodeling in the border and remote regions (i.e.: cardiomyocyte apoptosis, cardiomyocyte size, interstitial and perivascular fibrosis, capillary and arteriolar density).

3) The authors focused on male rats. This shold be highlighted in the title. Moreover, they should discuss the lack of female rats as a limitation of the study.

4) Add data on body weight and heart weight/body weight ratio, ejection fraction and LV diastolic function.

5) The authors should add RNA-seq dataset as supplementary file.

6) The authors should clarify data showed in Fig.4. In particular, the impact of genes allocated in Cluster 1. Indeed, previous studies on myocardial metabolism hase demonstrated the role of CPT1 and fatty acid metabolism in modulating contractile response and remodeling (please see Cardiovasc Res. 2005 Jun 1;66(3):454-61.; Nature 622, 619–626 (2023))

7) Figure legend should include the statistical analysis used for the showed data.

Reviewers' comments:

Reviewer's Responses to Questions

**Comments to the Author**

1. Is the manuscript technically sound, and do the data support the conclusions?

Reviewer #1: Partly

Reviewer #2: Partly

2. Has the statistical analysis been performed appropriately and rigorously?

Reviewer #1: Yes

Reviewer #2: No

3. Have the authors made all data underlying the findings in their manuscript fully available?

Reviewer #1: Yes

Reviewer #2: Yes

4. Is the manuscript presented in an intelligible fashion and written in standard English?

Reviewer #1: Yes

Reviewer #2: Yes

Reviewer #1: The manuscript entitled “Attenuating effects of inflammatory pathway by prolonged left ventricular unloading after myocardial infarction in rats” is a preclinical assessment of unloading in the setting AMI (LAD ligation) using a heterotopic heart-lung transplant model for partial unloading to determine inflammatory signaling. The primary finding is that partial unloading exerts effects on inflammatory gene expression and macrophage polarization. It is then suggested that inflammatory modulation might be mechanistically related to the cardioprotective effects of mechanical unloading. The manuscript is well-written, the topic is clinically relevant, and the analysis is robust however I have a few concerns that I think should be addressed.

My major concern is regarding the partial unloading model as well as the lack of adjudication of infarct size done in any of the animals. In Fig 2B, it appears that fibrosis was elevated in the unloaded animals compared to AMI. How was fibrosis quantified? This is not described in the methods. As far as the other cardiac functional parameters described in Figure 2, I do not see any major statistical differences between the AMI and AMI/UL animals, which makes me wonder about the degree of unloading and if this was significant enough to unmask inflammatory changes that would be expected with a transvalvular flow pump (as would be delivered in humans). Additionally, the claim that “Partial UL attenuates deterioration of cardiac function after AMI but exerts” needs to be restated, given that the functional data does not suggest any relative improvement in cardiac function with partial UL. There are also multiple mentions of the cardioprotective effects of unloading, however I do not see that infarct size was quantified in this study. Was TTC staining done? Or at least quantification of histologic scar area? Indeed, if infarct size were lower in the partial unloading arm, it would be expected that their cardiac functional parameters would also be different than the non-unloaded AMI patients.

Methods – How were areas defined - “remote region,” “non-infarcted area,” and “viable myocardium” are all used. This needs to be more clearly laid out in the methods (what samples were taken and where). Was sampling from remote areas standardized?

Methods – it appears that a two -way ANOVA was used to compare the 4 groups. However, I think I would consider utilizing the 3 groups (control, AMI, AMI/UL) as the results would be more clinically relevant. There is no biologic basis for using UL without an inciting event nor is it an appropriate control/sham arm either.

QUANTISEQ – it seems that this package was used for rat myocardium, however this is a computational pipeline from human RNQ-seq data. Please clarify this point, which is not mentioned in the manuscript.

Figure 1 – needs more descriptive legend. What do the white and black bars represent?

“Several studies have demonstrated that UL can reduce infarct size, preserve cardiac function, and modulate myocardial metabolic stress when initiated during ischemia or at the time of reperfusion [6-8]. Based on these findings, UL has been increasingly applied in clinical settings, such as the percutaneous Impella device [9, 10].” – Would rephrase this. UL for the purpose of cardioprotection is only investigational at the current time. The Door-to-Unload pivotal trial results investigating the use of Impella CP at the time of AMI (STEMI) have not been released. Otherwise, placing a mechanical circulatory support device (IABP or ECMO) at the time of AMI for the purpose of cardioprotection have both been investigated in randomized studies (CRISP-AMI and ECLS-SHOCK) but with negative results. UL has been increasingly utilized for cardiogenic shock and high risk PCI, however this is only to provide hemodynamic support and not to reduce infarct size. Would scale back these claims.

Reviewer #2: The manuscript examines the effects of prolonged partial left ventricular unloading (UL) on inflammation after acute myocardial infarction (AMI) in rats. It explores the impact of UL on inflammatory pathways, immune cell behavior, and macrophage polarization in non-infarcted myocardium. Through RNA sequencing, immune cell deconvolution, qPCR, and immunohistochemistry, the study reveals that UL reduces cardiac functional decline, inflammation, and M2 macrophage accumulation following AMI.

The emphasis on the remote area is a notable strength of the manuscript since most literature concentrates on the infarct zone and the border zone. However, there are significant concerns that require the authors' attention.

1. The functional and histopathological data presented are remarkably similar to those recently published by the authors in another paper (Int. J. Mol. Sci. 2025, 26, 4422. https://doi.org/10.3390/ijms26094422), which raises concerns about the originality of the findings. The authors should address and discuss this issue.

2. In studies involving vertebrates, it is essential to evaluate animal welfare and clearly define experimental endpoints. Methods should be reported in detail with transparency. Information and details unsuitable for the main manuscript can be included as supplementary material. Was the infarction induction via LAD ligation and heterotopic heart-lung transplantation contextual? Were the donor and recipient rats syngeneic? What precautions, if any, were taken to minimize rejection risk? Did the animals receive prophylactic or post-surgical treatments? What type of anesthesia was used during echocardiographic imaging sessions?

3. Given the small sample size of the groups and the high variability of many parameters analyzed, a non-parametric statistical analysis would probably be more appropriate.

4. I acknowledge that comparing ultrasound results between UL and non-UL is impractical due to the differing hemodynamic conditions involved. This issue likely renders the statistical analysis presented in the table rather meaningless.

5. The presentation of results often appears biased, as it fails to address the impact of heterotopic heart–lung transplantation. For instance, the rate of fibrosis in UL samples must be considered, regardless of AMI, along with the increase in some cytokines.

6. In the RNAseq data, there appears some variability among the AMI groups with and without UL. What are the characteristics of these animals regarding the uniformity of infarct size and, at least for those without UL, also concerning ultrasound-derived cardiac function parameters?

7. In Figure 5, examining the histograms, 5 out of 11 genes show increased expression after AMI. I believe it is significant that UL alone causes an increase in expression, and that it is then minimally affected by AMI. A discussion on this point by the authors is requested. Furthermore, I find it confusing to use those colors to indicate the noUL and UL groups, especially since those colors also represent the z-score in the heatmaps (which is, incidentally, on the y-axis).

8. The use of deconvolution tools in non-tumor samples has been employed, often with similar results. However, it requires careful interpretation of fractions in normal or mixed tissues when tumor populations are absent. The authors are encouraged to address this issue.

9. Figure 8 shows quantitative analysis of iNOS immunohistochemistry data to represent M1 class of macrophages, but does not include any representative micrograph.

10. The idea that inflammation following an acute myocardial infarction (AMI) extends beyond the infarcted area into non-infarcted regions—and that this may contribute to global ventricular function impairment and remodeling—is a significant yet relatively novel concept in cardiology. However, it is not original. Several studies, both in relevant preclinical models and in humans, appear to have established this connection. An illustrative example is the study titled "Inflammation in Remote Myocardium and Left Ventricular Remodeling After Acute Myocardial Infarction: A Pilot Study Using T2 Mapping." You can find it at the following link: https://doi.org/10.1002/jmri.27827.

11. The conclusion presented in the abstract appears to contradict the conclusion found in the main text. The abstract states that partial unloading (UL) attenuates the deterioration of cardiac function but has limited effects on inflammatory gene expression and macrophage polarization in the non-infarcted area. This suggests that the cardioprotective effect is only partially linked to inflammation in the remote area, with complex influences stemming from fibrosis and atrophy. In contrast, the second paragraph claims that partial UL significantly affects inflammatory gene expression and macrophage polarization in the remote area, implying that cardiac protection results from a more pronounced modulation of inflammatory pathways in that region. The authors need to address this issue.

**Do you want your identity to be public for this peer review?** For information about this choice, including consent withdrawal, please see our Privacy Policy

Reviewer #1: No

Reviewer #2: No

---

## [Author Response · Author response to Decision Letter 1]

7 Jan 2026

Responses to the additional editor comments:

First of all, we sincerely thank the editor for all the valuable comments to improve our manuscript.

Comments to the Authors:

1) The manuscript provides valuable insights into post-infarction remodeling and the effects of LV unloading but fails to consider the potential involvement of the canonical Wnt/β-catenin pathway, a key regulator of inflammation and fibrosis after MI (please see Cardiovasc Res, 2016;112:645–655). Given the overlap between the authors’ findings and Wnt-regulated processes, this omission limits the mechanistic depth of the study. The authors should analyze canonical Wnt pathway activation using their existing RNA-seq dataset (i.e.: KEGG/GSEA enrichment or marker gene expression such as CTNNB1, AXIN2, LRP6, TCF7L2) to strengthen the biological interpretation. Finally, they should discuss the results in the light of abovementioned study.

Author’s response:

We appreciate the editor’s suggestion regarding canonical Wnt/β-catenin signaling. To address this point, we evaluated the related KEGG pathway in different groups from RNAseq, and Wnt/β-catenin signaling was not detected (see the figures below). Also, the expressions of four representative canonical Wnt markers (Ctnnb1, Axin2, Lrp6, Tcf7l2) were analyzed, and none of these genes showed significant differences among the four groups. Furthermore, no MI × unloading interaction was detected (Ctnnb1: p = 0.35, Axin2: p = 0.15, Lrp6: p = 0.90, Tcf7l2: p = 0.68).

These data suggest that canonical Wnt signaling is not prominently activated in the remote myocardium in this model, and therefore additional pathway-level analyses would likely not yield mechanistically informative results. Nevertheless, we have added a brief discussion on the potential relevance of Wnt signaling and its reported involvement in post-MI remodeling: “Canonical Wnt/β-catenin signaling has been implicated in post-MI inflammation and fibrotic remodeling in previous studies [20, 21]. However, in the present study, the expression of representative canonical Wnt markers (Ctnnb1, Axin2, Lrp6, and Tcf7l2, data not shown) was not significantly altered among groups, and no interaction between MI and unloading was detected. These data suggest that canonical Wnt activation was not prominent in the remote myocardium under our experimental conditions, and that the inflammatory modulation observed here may be governed by other signaling mechanisms.” (line 452-460).

Comments to the Authors:

2) The authors should complete the histological characterization of hallmarks of myocardial remodeling in the border and remote regions (i.e.: cardiomyocyte apoptosis, cardiomyocyte size, interstitial and perivascular fibrosis, capillary and arteriolar density).

Author’s response:

We appreciate the editor’s suggestion. We added cardiomyocyte size, interstitial and perivascular fibrosis analysis in the border zone (see Fig. S2, line 253-254). Cardiomyocyte apoptosis and capillary and arteriolar density were not assessed in the present study because their accurate evaluation requires specialized experimental approaches and marker-based histological methods, which were not included in the original study design. We have clarified this point as a limitation of the study: “Finally, we did not quantify infarct size, cardiomyocyte apoptosis, and capillary and arteriolar density in this study. Because only a single papillary-level section was collected per heart, and specialized staining is necessary, reliable estimation was not feasible” (line 526-532).

Comments to the Authors:

3) The authors focused on male rats. This should be highlighted in the title. Moreover, they should discuss the lack of female rats as a limitation of the study.

Author’s response:

We thank the editor for this comment. The current study used male rats to minimize variability associated with sex-specific hormonal cycles and because our primary aim was to characterize inflammatory and macrophage responses under defined experimental conditions. We have added male rats in the title and a statement in the limitations section acknowledging the lack of female animals and noting that future studies are warranted to determine whether the effects of unloading differ between sexes: “Third, only male rats were used, which may limit generalizability, and whether sex modifies the effects of unloading after MI should be evaluated in the future.” (line 525-526).

Comments to the Authors:

4) Add data on body weight and heart weight/body weight ratio, ejection fraction and LV diastolic function.

Author’s response:

We thank the editor for this suggestion. We have now included data on body weight, LV weight, and LV weight/body weight ratio in a new table (Supplementary Table S3.). These parameters are presented as mean ± SEM across the four experimental groups (non-AMI, AMI, non-AMI/UL, and AMI/UL). In addition, we have added the ejection fraction (EF) values by the Teichholz method at baseline and at 14 days post-surgery (Supplementary Table S4.). Diastolic function was not assessed due to little LV inflow in the UL group.

Comments to the Authors:

5) The authors should add RNA-seq dataset as supplementary file.

Author’s response:

We appreciate the editor’s recommendation. The RNA-seq dataset was deposited in a publicly accessible database, and the corresponding access link is included in the revised manuscript: “RNA-seq was performed by Takara Bio Inc. (Kanagawa, Japan), and the data were uploaded into Annotare 2.0 with an accession number of E-MTAB-16419.” (line 154-156).

Comments to the Authors:

6) The authors should clarify data showed in Fig.4. In particular, the impact of genes allocated in Cluster 1. Indeed, previous studies on myocardial metabolism have demonstrated the role of CPT1 and fatty acid metabolism in modulating contractile response and remodeling.

Author’s response:

We appreciate the editor’s comment. To address this point, we analyzed the expression of CPT1, CD36, and Fabp3 family genes within Cluster 1. We revised the explanation in the results: “Genes in cluster 1 were related to fatty acid metabolism, oxidative phosphorylation, adipogenesis, peroxisome, and bile acid metabolism. The expression of Cpt1b, CD36, and Fabp3, which are related to fatty acid metabolism, was significantly decreased after AMI. In cluster 2, the genes were mainly related to epithelial–mesenchymal transition, TNFα signaling via NF-κB, allograft rejection, inflammatory response, and IL-6–JAK–STAT3 signaling (Fig. 4B)”, (line 288-293).

Comments to the Authors:

7) Figure legend should include the statistical analysis used for the showed data.

Author’s response:

We thank the reviewer for this comment. We have revised the figure legends to include the statistical analysis used for each figure, including the type of statistical test, post-hoc comparisons (where applicable), and sample sizes. This information is now clearly stated in the legends for Figures.

Reviewer #1:

First of all, we sincerely thank the Reviewer #1 for all the valuable comments to improve our manuscript.

Comments to the Authors:

1) My major concern is regarding the partial unloading model as well as the lack of adjudication of infarct size done in any of the animals. In Fig 2B, it appears that fibrosis was elevated in the unloaded animals compared to AMI. How was fibrosis quantified? This is not described in the methods.

Author’s response:

We appreciate your valuable comments. We were unable to quantify infarct size in this study, as only a single papillary-level section was collected per heart. We also apologize for the lack of methodological detail in the previous version. We added this in the method section as follows: “Fixed LV tissue blocks were sectioned and stained with hematoxylin–eosin to assess myocyte morphology. Cardiomyocyte cross-sectional area was quantified in non-infarcted myocardium by measuring 100 randomly selected myocytes containing a visible central nucleus per animal. Areas were outlined and calculated using ImageJ (NIH, 1.54f, U.S. National Institutes of Health; Bethesda, MD, USA). Masson’s trichrome staining was employed to determine the extent of fibrosis. For each animal, ten random microscopic fields from the non-infarcted zone were imaged, and collagen-rich regions were segmented using threshold-based detection in ImageJ 1.54f. Fibrosis was expressed as the percentage of stained area relative to total tissue area, and mean values per animal were used in group-level analyses.” (line 139-149).

Comments to the Authors:

2) As far as the other cardiac functional parameters described in Figure 2, I do not see any major statistical differences between the AMI and AMI/UL animals, which makes me wonder about the degree of unloading and if this was significant enough to unmask inflammatory changes that would be expected with a transvalvular flow pump (as would be delivered in humans).

Author’s response:

We appreciate your valuable comments. AMI/UL was performed by heterotopic heart-lung transplantation after AMI; thus, it was unable to be directly compared with the AMI group, which did not receive transplantation. We try to solve this issue by adapting two-way ANOVA, with two other sham groups (non-AMI and non-AMI/UL), and we explain this in comment 5.

As you can see in Supplementary Table S4., 14 d LVEDD and LVESD were almost half of the values of baseline, which suggests a significant LV unloading.

Comments to the Authors:

3) Additionally, the claim that “Partial UL attenuates deterioration of cardiac function after AMI but exerts” needs to be restated, given that the functional data does not suggest any relative improvement in cardiac function with partial UL. There are also multiple mentions of the cardioprotective effects of unloading, however I do not see that infarct size was quantified in this study. Was TTC staining done? Or at least quantification of histologic scar area? Indeed, if infarct size were lower in the partial unloading arm, it would be expected that their cardiac functional parameters would also be different than the non-unloaded AMI patients.

Author’s response:

We thank the reviewer for this important comment. We agree that the original phrasing may have overstated the functional impact of unloading. In the revised manuscript, we stated as follows: “Partial UL suggests a potential influence on cardiac function after AMI but exerts effects on inflammatory gene expression and macrophage polarization in the non-infarcted area” (line 535-537). We are sorry that we were unable to quantify infarct size using the available samples. Because only a single papillary-level section was collected from each heart to evaluate the non-infarcted myocardium, quantifying infarct size from these samples would be biased and unreliable. Accurate assessment of infarct size requires serial sectioning throughout the ventricle or whole-heart TTC staining, which was not included in our experimental protocol. We acknowledge this limitation and have now stated it explicitly in the revised manuscript: “Future studies will incorporate serial histological sectioning or whole-heart TTC staining to accurately determine infarct size and evaluate whether unloading affects acute myocardial injury.” (line 529-532).

Comments to the Authors:

4) Methods – How were areas defined - “remote region,” “non-infarcted area,” and “viable myocardium” are all used. This needs to be more clearly laid out in the methods (what samples were taken and where). Was sampling from remote areas standardized?

Author’s response:

We apologize for the inconvenience. We revised the methods section and added the identification for the areas as follows: “The infarct area was identified as the macroscopically white region. The border area was defined as the area extending 3 mm from the infarct area to the free wall. The remaining zone was a remote area.” (line 134-136).

Comments to the Authors:

5) Methods – it appears that a two -way ANOVA was used to compare the 4 groups. However, I think I would consider utilizing the 3 groups (control, AMI, AMI/UL) as the results would be more clinically relevant. There is no biologic basis for using UL without an inciting event nor is it an appropriate control/sham arm either.

Author’s response:

We appreciate the reviewer’s suggestion. A direct comparison among only three groups (control, AMI, and AMI+UL) would not allow us to distinguish the intrinsic effects of unloading (UL) from its modulatory effect on AMI. Because in the AMI/UL group, UL is implemented via a transplantation-based model, it inevitably introduces biological effects independent of unloading, and therefore, it is inappropriate to compare AMI and AMI/UL groups directly.

For example, in experimental biology, a knockout (KO) model involves inactivating a specific gene to examine its function. KO studies commonly use a four-group design—wild-type controls, wild-type with acute myocardial infarction (AMI), KO controls, and KO with AMI—to test whether gene deletion modifies the response to AMI through evaluation of the interaction term. Analogously, in the present study, LV unloading (UL) was treated as the modifying factor, resulting in four experimental groups (non-AMI, AMI, non-AMI/UL, and AMI/UL). The data were analyzed using two-way ANOVA with AMI and UL as factors. This framework enables explicit evaluation of the interaction term, which directly tests whether UL modifies the biological response to AMI beyond its baseline effects. A significant interaction, therefore, addresses our primary biological question and avoids potentially misleading conclusions that could arise from pairwise comparisons alone. This approach is methodologically standard in genetic and pharmacological interaction studies and allows valid inference despite the unavoidable secondary effects associated with the UL model.

Comments to the Authors:

6) QUANTISEQ – it seems that this package was used for rat myocardium, however this is a computational pipeline from human RNQ-seq data. Please clarify this point, which is not mentioned in the manuscript.

Author’s response:

We thank the reviewer for pointing out an important methodological concern. We acknowledge that QUANTISEQ was originally developed and validated using human RNA-seq data, and we adopted it in this study because: (1) comparable deconvolution tools validated for rat cardiac tissue are currently lacking; (2) the bulk RNA-seq data from our rat myocardium were of high quality, and immune-cell infiltration and inflammatory changes were among our key hypotheses; (3) using a human-derived signature matrix represents an attempt to infer immune cell composition, under the assumption that many immune cell–specific marker genes are conserved across mammals. Nevertheless, we fully agree that this represents a limitation. Also, we performed immunostaining for macrophage markers, which confirmed increased M2 macrophage infiltration predicted by the computational analysis, providing independent support for this component of the RNA-seq findings. We added some explanation in the method section as follows: “Although QUANTISEQ was originally developed for human RNA-seq data, we applied it to rat myocardial RNA-seq because no rat-specific immune deconvolution pipelines are currently available and many immune-cell signature genes are evolutionarily conserved across species. We used the default human TIL10 signature matrix and performed deconvolution on normalized TPM data from our bulk RNA-seq.” (line 174-178).

Comments to the Authors:

7) Figure 1 – needs more descriptive legend. What do the white and black bars represent?

Author’s response:

We thank the reviewer for the comment. The legend for Figure 1 has been revised to provide clearer descriptions of the data presentation. Specifically, we now indicate what the white and black bars represent: “White bars denote periods without intervention; black bars represent periods with intervention (LAD ligation or unloading).” (line 109-111).

Comments to the Authors:

8) “Several studies have demonstrated that UL can reduce infarct size, preserve cardiac function, and modulate myocardial metabolic stress when initiated during isc

---

## [Decision Letter · Decision Letter 1]

26 Jan 2026

Dear Dr. Shingu,

Thank you for submitting your manuscript to PLOS ONE. After careful consideration, we feel that it has merit but does not fully meet PLOS ONE’s publication criteria as it currently stands. Therefore, we invite you to submit a revised version of the manuscript that addresses the points raised during the review process.

**ACADEMIC EDITOR:** Despite some efforts by the authors, relevant issues need to be solved in accord with Editor's and Reviewers' suggestions. These issues are required.

We look forward to receiving your revised manuscript.

Kind regards,

Vincenzo Lionetti, M.D., PhD

Academic Editor

PLOS One

Journal Requirements:

Reviewers' comments:

Reviewer's Responses to Questions

**Comments to the Author**

Reviewer #2: (No Response)

2. Is the manuscript technically sound, and do the data support the conclusions?

Reviewer #2: Yes

3. Has the statistical analysis been performed appropriately and rigorously?

Reviewer #2: Yes

4. Have the authors made all data underlying the findings in their manuscript fully available?

Reviewer #2: Yes

5. Is the manuscript presented in an intelligible fashion and written in standard English?

Reviewer #2: Yes

Reviewer #2: 1. Even after the authors' response to the matter, which I consider only partially satisfactory, I continue to believe that estimating infarct size would have given the results greater weight and perhaps even dispelled some doubts about the infarct's outcome and the variability between the groups. At least macroscopic morphometry of the infarcted region in fresh tissue (preferably, though not necessarily, using TTC), given that the authors identified it as the "white region." This limitation affects the manuscript's overall impact.

2. In Table S4, ESD unexpectedly decreased by about 40% from baseline at 14 days post-AMI in non-unloaded rats, in contrast to the typical progressive dilation. This atypical pattern alongside EDD changes requires methodological verification and discussion.

3. The authors' response to my comment #7 is partially satisfactory for the following reasons: a) Dismissing the confusion regarding the color scheme as "automatically generated and uneditable" is unconvincing. A good scientific publication requires clear visualizations controlled by the authors, where effective scientific communication takes precedence over software limitations. b) Despite the reviewer's explicit request, there is no discussion paragraph that addresses the biological significance of the effects of UL/surgery on these genes, leaving a crucial observation unexplained.

4. The "major concern" regarding the partial unloading model—specifically mentioned in my comment #5 and reviewer #1's comment #1—has not been addressed. There is no explanation provided for model validation, unloading efficacy metrics, or the rationale for the unexpected increase in fibrosis in the unloaded groups compared to the AMI groups. This is counterintuitive, given that the intent of unloading is to have an anti-fibrotic effect.

**Do you want your identity to be public for this peer review?** For information about this choice, including consent withdrawal, please see our Privacy Policy

Reviewer #2: No

---

## [Author Response · Author response to Decision Letter 2]

2 Feb 2026

Reviewer #2:

First of all, we sincerely thank Reviewer #2 for all the valuable comments to improve our manuscript.

Comments to the Authors:

1. Even after the authors' response to the matter, which I consider only partially satisfactory, I continue to believe that estimating infarct size would have given the results greater weight and perhaps even dispelled some doubts about the infarct's outcome and the variability between the groups. At least macroscopic morphometry of the infarcted region in fresh tissue (preferably, though not necessarily, using TTC), given that the authors identified it as the "white region." This limitation affects the manuscript's overall impact.

Author’s response:

We thank the reviewer for the comment. To address this concern, we attempted to use Masson’s trichrome staining to estimate the fibrotic area in the UL groups. However, in AMI/UL samples, intraventricular thrombus was present and tightly attached to the infarcted myocardium. Since both collagen-rich infarct tissue and thrombus are stained blue by Masson’s trichrome, it was not possible to reliably distinguish the infarct core from the thrombus or surrounding tissue. Therefore, accurate and reproducible delineation of infarct borders could not be achieved.

Regarding TTC staining, as the reviewer mentioned, fresh tissue is required. Since all samples in the present study had already been collected and processed for downstream molecular and histological analyses, TTC staining could not be retrospectively performed. Even if the AMI and UL surgeries were repeated, the newly obtained samples would not be directly comparable to those used in the current experiments. Therefore, TTC staining was not pursued in this study and the present results are based on the existing experimental dataset.

Comments to the Authors:

2. In Table S4, ESD unexpectedly decreased by about 40% from baseline at 14 days post-AMI in non-unloaded rats, in contrast to the typical progressive dilation. This atypical pattern alongside EDD changes requires methodological verification and discussion.

Author’s response:

We sincerely thank the reviewer for this careful observation, and we apologize for the confusion caused by an error in Table S4. After receiving this comment, we rechecked our laboratory records and confirmed that the LVESD values at day 14 in all groups had been transcribed incorrectly. Specifically, during data transfer to the spreadsheet, a wall thickness parameter was mistakenly entered in place of LVESD, which artificially produced an apparent decrease in ESD after AMI. We have now corrected the LVESD dataset using the original measurement files, updated Table S4 accordingly, and re-ran all downstream calculations and statistical analyses that used these values (including EDD/ESD-derived indices). With the corrected data, the post-AMI pattern is consistent with expected ventricular remodeling, and the atypical ESD decrease is no longer present.

Comments to the Authors:

3. The authors' response to my comment #7 is partially satisfactory for the following reasons: a) Dismissing the confusion regarding the color scheme as "automatically generated and uneditable" is unconvincing. A good scientific publication requires clear visualizations controlled by the authors, where effective scientific communication takes precedence over software limitations. b) Despite the reviewer's explicit request, there is no discussion paragraph that addresses the biological significance of the effects of UL/surgery on these genes, leaving a crucial observation unexplained.

Author’s response:

We thank the reviewer for this comment.

a) In the revised manuscript, we have adjusted the color scheme of the MultiDEG figure (Fig. 5A) to improve clarity and to make it clearly distinguishable from Fig. 4B.

b) In addition, we have added a new paragraph in the Discussion section to address the biological significance of the effects of UL/surgery on these genes. “Among these inflammation-related hub genes, Il1r2, Il2ra, and RT1-Db1 exhibited significant AMI × UL interaction effects, suggesting that their expression is specifically modulated by UL in post infarction. Notably, previous clinical evidence has shown that circulating Il1r2 levels in patients with myocardial infarction are independently associated with parameters of adverse left ventricular remodeling [20]. In our study, UL alone caused an increase in Il1r2 expression, and it was minimally affected by AMI. Thus, the suppression of Il1r2 in the AMI/UL group may possibly reflect a shift in inflammatory signaling toward a transcriptional profile associated with less adverse remodeling, although causality cannot be inferred from the present data. No studies have directly linked Il2ra or RT1-Db1 to AMI or mechanical unloading. Therefore, our data provide novel evidence that ventricular UL modulates inflammation-related gene programs in the remote myocardium after AMI, highlighting a previously underexplored interaction between UL and post-AMI inflammatory regulation.” (line 448-461).

Comments to the Authors:

4. The "major concern" regarding the partial unloading model—specifically mentioned in my comment #5 and reviewer #1's comment #1—has not been addressed. There is no explanation provided for model validation, unloading efficacy metrics, or the rationale for the unexpected increase in fibrosis in the unloaded groups compared to the AMI groups. This is counterintuitive, given that the intent of unloading is to have an anti-fibrotic effect.

Author’s response:

We appreciate the reviewer’s concern regarding the validation of the partial unloading model. The model validation and unloading efficacy have been demonstrated by the echocardiographic parameters, as described in question 2 (supplementary table S4), which confirm the effective reduction of mechanical LV load in the UL groups.

We would like to clarify that the fibrosis data presented in this study represent interstitial fibrosis in the remote LV myocardium, rather than fibrosis within the infarct zone or global myocardial fibrosis. Therefore, these results should not be interpreted as infarct size or total scar formation. In the present experimental setting, accurate quantification of fibrosis within the infarct region was not feasible due to the presence of intraventricular thrombus and hemorrhagic components, which were co-stained with collagen in Masson’s trichrome sections, as explained in question #1.

In addition, there have been studies reporting that fibrosis increased after LV unloading. Bello SO et al. performed unloading in a rat AMI model, and the results showed that fibrosis increased in the UL group (Bello SO et al., Mechanical unloading coupled with coronary reperfusion stimulates cardiomyocyte proliferation and prevents unloading-induced fibrosis after myocardial infarction. Basic Res Cardiol. 2025). In a study using patient data after LVAD, it was also reported that LVAD support increases LV collagen cross-linking and the ratio of collagen type I to III, as well as myocardial stiffness (Klotz S et al., Mechanical unloading during left ventricular assist device support increases left ventricular collagen cross-linking and myocardial stiffness. Circulation. 2005).

---

## [Decision Letter · Decision Letter 2]

8 Feb 2026

Dear Dr. Shingu,

Thank you for submitting your manuscript to PLOS ONE. After careful consideration, we feel that it has merit but does not fully meet PLOS ONE’s publication criteria as it currently stands. Therefore, we invite you to submit a revised version of the manuscript that addresses the points raised during the review process.

**ACADEMIC EDITOR: The authors shoul edit reference #21 in accord with Editor's suggestion**
**==============================**

We look forward to receiving your revised manuscript.

Kind regards,

Vincenzo Lionetti, M.D., PhD

Academic Editor

PLOS One

Journal Requirements:

Additional Editor Comments:

Reference #21 is not appropriated. Regarding the relationship between Wnt and MI the suggested reference was Cardiovasc Res. 2016 Dec;112(3):645-655. doi: 10.1093/cvr/cvw214.  Please edit.

Reviewers' comments:

Reviewer's Responses to Questions

**Comments to the Author**

Reviewer #2: All comments have been addressed

2. Is the manuscript technically sound, and do the data support the conclusions?

Reviewer #2: (No Response)

3. Has the statistical analysis been performed appropriately and rigorously?

Reviewer #2: (No Response)

4. Have the authors made all data underlying the findings in their manuscript fully available?

Reviewer #2: (No Response)

5. Is the manuscript presented in an intelligible fashion and written in standard English?

Reviewer #2: (No Response)

Reviewer #2: The authors have adequately addressed my comments raised in a previous round of review. The manuscript is now improved.

Do you want your identity to be public for this peer review? For information about this choice, including consent withdrawal, please see our Privacy Policy

Reviewer #2: No

---

## [Author Response · Author response to Decision Letter 3]

8 Feb 2026

Additional Editor Comments:

First of all, we sincerely thank the editor for the valuable comment to improve our manuscript.

1. Reference #21 is not appropriate. Regarding the relationship between Wnt and MI the suggested reference was Cardiovasc Res. 2016 Dec;112(3):645-655. doi: 10.1093/cvr/cvw214. Please edit.

Author’s response:

We thank the editor for the comment. We apologize for any inconvenience. We have revised the reference.

---

## [Editor Report · Decision Letter 3]

10 Feb 2026

Attenuating effects of inflammatory pathway by prolonged left ventricular unloading after myocardial infarction in male rats

PONE-D-25-55622R3

Dear Dr. Shingu,

We’re pleased to inform you that your manuscript has been judged scientifically suitable for publication and will be formally accepted for publication once it meets all outstanding technical requirements.

Kind regards,

Vincenzo Lionetti, M.D., PhD

Academic Editor

PLOS One
---

## [Editor Report · Acceptance letter]

PONE-D-25-55622R3

PLOS One

Dear Dr. Shingu,

I'm pleased to inform you that your manuscript has been deemed suitable for publication in PLOS One. Congratulations! Your manuscript is now being handed over to our production team.

Kind regards,

on behalf of

Prof. Vincenzo Lionetti

Academic Editor

PLOS One